# New Pamphiliids with Varying Venations from Lower Cretaceous Yixian Formation of Northeast China (Hymenoptera, Pamphiliidae) [note 1]

**DOI:** 10.3390/insects13100947

**Published:** 2022-10-18

**Authors:** Jialiang Zhuang, Alexandr P. Rasnitsyn, Chungkun Shih, Dong Ren, Mei Wang

**Affiliations:** 1Key Laboratory of Forest Protection of National Forestry and Grassland Administration, Ecology and Nature Conservation Institute, Chinese Academy of Forestry, Beijing 100091, China; 2College of Life Sciences, Capital Normal University, 105 Xisanhuanbeilu, Haidian District, Beijing 100048, China; 3Paleontological Institute, Russian Academy of Sciences, 123, Profsoyuznaya ul., 117647 Moscow, Russia; 4Department of Palaeontology, Natural History Museum, Cromwell Road, London SW7 5BD, UK; 5Department of Paleobiology, National Museum of Natural History, Smithsonian Institution, Washington, DC 20013, USA

**Keywords:** Juralydinae, Sandaogou, venation variation, fossils, taxonomy

## Abstract

**Simple Summary:**

The presence of various morphologies of the ap-Cu, an extra vein with different lengths in the forewing, is a special character in Pamphiliidae. In Juralydinae, this vein is usually absent or extremely short. Here, we describe one new genus and three new species of Juralydinae with varying lengths of the ap-Cu from the Lower Cretaceous Yixian Formation in Duolun County, Inner Mongolia, China, and revise the diagnostic characteristics of Juralydinae. The new specimens expand the diversity of Pamphiliidae in the Mesozoic and strengthen our knowledge of venation variation.

**Abstract:**

One new genus and three new species of Pamphiliidae, *Dolicholyda obtusata* gen. et sp. nov., *Dolicholyda confluens* sp. nov., and *Dolicholyda angusta* sp. nov. are described and illustrated. All of them were collected from the Lower Cretaceous Yixian Formation in Duolun County, Inner Mongolia, China. The new genus is established based on the following characters: body surface without punctations; forewing with pterostigma lanceolate and sclerotized around the margins; angle between 1-M and 1-Cu nearly 90°; cell 1mcu long and obviously longer than length of pterostigma. In most cases, the ap-Cu is present, and its length varied. Additionally, we revise the diagnostic characteristics of Juralydinae based on the new specimens. New findings enhance our understanding of the wing venation characteristics of fossil pamphiliids and expand the diversity of Pamphiliidae in the Mesozoic.

## 1. Introduction

Pamphilioidea, one of the oldest and basal superfamilies of Hymenoptera, consist of four families: Mirolydidae, Xyelydidae, Pamphiliidae and Megalodontesidae [1,2,3,4]. Its earliest fossil records are dated to the Early Jurassic [5,6]. As one of the relict families of this superfamily, Pamphiliidae also have fossil records in the late Middle Jurassic to early Late Jurassic age [7,8,9]. Currently, two extant subfamilies, Pamphiliinae Cameron 1890 and Cephalciinae Benson 1945, and an extinct subfamily, Juralydinae Rasnitsyn 1977, are attributed to Pamphiliidae [10].

However, the extinct subfamily Juralydinae was not initially recognized after its establishment because of the lack of important information for identification and diagnosis [11,12]. Extant taxa of Pamphiliidae have been typically identified by tiny and rarely preserved characters, such as the configuration of the pre-apical spur in fore tibia, the inner tooth in the tarsal claw, the body surface colour, etc., and little attention was paid to wing venations [13,14]. However, the classifications of fossil Pamphiliidae have mainly focused on the structures often preserved, especially wing venation and antennae. The contradictions in the focal characters used for classification and the limited number of fossil specimens make it challenging to compare and classify fossil vs. extant pamphiliids. Therefore, some distinguishing features of Juralydinae still need to be deliberated. Jouault et al. [15] transferred *Tapholyda caplani* (Cockerell, 1933) (from Oligocene and Miocene) to Juralydinae and revised the diagnoses of Juralydinae. So far, the subfamily Juralydinae includes all the fossil species except *Acantholyda erythrocephala* (Linnaeus, 1758) (from Miocene), possibly *Acantholyda ribesalbesensis* Peñalver and Arillo, 2002 (from Miocene) and *Ulteramus republicensis* Archibald and Rasnitsyn, 2015 (from lower Eocene and its subfamily is undetermined) [16,17,18,19].

Herein, we describe three new species in a new genus *Dolicholyda* gen. nov. in Juralydinae, from the Lower Cretaceous Yixian Formation, and revise the diagnoses of Juralydinae again. In addition, we summarize and discuss wing venations and their variations for extant and extinct pamphiliids.

## 2. Materials and Methods

The fossil specimens of the new genus *Dolicholyda* gen. nov. are housed in the Key Laboratory of Insect Evolution and Environmental Changes, College of Life Sciences, Capital Normal University, Beijing, China (CNUB; Dong Ren, Curator). All of them were collected from the Lower Yixian Formation, Sandaogou, Nanyingpan Village, Duolun County, Inner Mongolia, China. The extant specimens of Pamphiliidae used for wing venation photos are deposited in the Natural History Museum of the Chinese Academy of Forestry.

The stratigraphic attribution of the Sandaogou locality has been controversial. It was assigned to the Sandaogou, Jianchang, Duolun, Jiufotang Formations successively, and eventually classified into the Lower Yixian Formation [20,21]. However, the precise age of Sandaogou in Duolun County has been unknown for a long time [22]. This area, as a member of the Yixian Formation, consists of two rock members: the lower one (K1y1) is a set of acidic volcanic-sedimentary rock strata, and the upper one (K1y2) is a set of trachytic volcanic rock strata. In 2019, the LA-ICP-MA zircon U-Pb ages of (132.2 ± 0.6) Ma and (132.8 ± 0.8) Ma were obtained in the rhyolite of the K1y1 rock member by Xue et al. [23]. Considering that the sedimentary rocks in this area exist in K1y1, we tentatively treat 132 Ma as the age of the fossil material in this study, which matches the period of the Jehol Biota (135–120 Ma) [24,25,26,27].

The specimens were examined and photographed under a Nikon SMZ 25 stereo microscope with an attached Nikon DS-Ri2 digital camera system, either dry or wetted with 95% ethanol. Wing venation terminology is adapted from Huber and Sharkey [28]. For wing venations, 1-Rs, 2-Rs, 3-Rs, and 4-Rs mean, respectively, the 1st, 2nd, 3rd, and 4th segments of Rs; 1-M and 2-M mean, respectively, the 1st and 2nd segments of M; 1-Cu, 2-Cu, and 3-Cu refer, respectively, to the 1st, 2nd, and 3rd segments of Cu. We treat the extra vein located at the turning of M+Cu as “ap-Cu”, which follows Archibald and Rasnitsyn [19].

It is worth noting that wings of specimens are broken at the locations of the anal fold and the jugal fold, for example, the location near vein 1A in the hind wing of specimen CNU-HYM-ND2022105p/c, and vein 1A in both forewings of specimen CNU-HYM-ND2022108. We have deliberately reconstructed the wing venations with reference to other fossil specimens in the line drawing pictures.

## 3. Results

Systematic Palaeontology.

Order Hymenoptera Linnaeus, 1758.

Suborder Symphyta Gerstaecker, 1867.

Superfamily Pamphilioidea Cameron, 1890.

Family Pamphiliidae Cameron, 1890.

Subfamily Juralydinae Rasnitsyn, 1977.

Type genus. *Juralyda* Rasnitsyn, 1977.

Diagnosis (modified after Jouault et al., 2022, [15]). Antenna with first flagellomere nearly as long as next three flagellomeres and distinctly thicker than these (also in some Cephalciinae, e.g., *Caenolyda* Konow, 1897). Forewing with Sc2 if present/preserved meeting R before 1-Rs (also in all pamphiliid genera except *Ulteramus* Archibald and Rasnitsyn, 2016 [19]). 1-Rs about half as long as, or slightly longer than, 1-M (the only uniquely diagnostic feature of the subfamily). M+Cu angular with or without ap-Cu of various length at elbow (uniquely diagnostic feature of the family Pamphiliidae except that M+Cu is gently bent and lacking ap-Cu in *Tapholyda* Rasnitsyn, 1983, [15]). Cell 1mcu usually shorter than length of pterostigma (except *Dolicholyda* gen. nov.).

Included genera. *Juralyda* Rasnitsyn, 1977; *Scabolyda* Wang, Rasnitsyn, Shih and Ren, 2014; *Atocus* Scudder, 1892; *Tapholyda* Rasnitsyn, 1983 and *Dolicholyda* gen. nov.

Remarks. The present addition of *Dolicholyda* gen. nov. to Juralydinae affects considerably the diagnosis of the subfamily as outlined by Jouault et al. [15]. A long 1-Rs is left the only uniquely diagnostic feature of the subfamily, with the diagnostic meaning of others (given in quotation marks below) changed by this additional genus, or for other reasons as follows:‘First flagellomere is distinctly thicker than following flagellomeres, twice to three times as long as second’: the same holds true for *Caenolyda* (Cephalciinae); not thicker and less long in *Atocus* [29].‘M+Cu gently bent, with ap-Cu extremely short when present’: this is correct for *Tapholyda* only, suggesting a secondary loss of the M+Cu angulation in that genus.‘cell 1mcu shorter than length of pterostigma’: incorrect for *Dolicholyda* gen. nov.‘cell 2r much wider than cell 2r-m’: the feature is characteristic of the vast majority of Pamphiliidae.‘3r-m separated from 2m-cu by a distance equal or subequal to its length’: the feature is characteristic of the vast majority of Pamphiliidae.

Genus: *Dolicholyda* Zhuang, Rasnitsyn, Shih and Wang, gen. nov.

Etymology: The generic name is a combination of Greek “*dolich-*”, meaning long and referring to the long and narrow cell 1mcu, and the generic name *Lyda* (often used in Pamphiliidae). Gender: feminine.

Type species. *Dolicholyda obtusata* Zhuang, Rasnitsyn, Shih and Wang, sp. nov.

Diagnosis. Body surface without punctations. Length-width ratio of flagellomeres less than 3:1 excluding first flagellomere. Forewing with pterostigma lanceolate, not sclerotized inside and sclerotized along margins; Sc developed and bifid, Sc2 merging into R anteriad 1-Rs; Rs+M at least 1.5 times as long as 2-M; angle between 1-M and 1-Cu nearly 90°; cell 1r nearly twice as long as wide; cell 1mcu rectangular, long and narrow, and obviously longer than length of pterostigma; 3r-m inclined toward the wing apex. In most cases, ap-Cu present and of variable length.

Remarks. *Dolicholyda* gen. nov. is assigned to Pamphiliidae based on the combination of forewing with Sc developed; M+Cu angularly bent; antenna with pedicel not inflated, first flagellomere mostly 2–3 times as long as the second. The new genus matches the main diagnostic character of Juralydinae, forewing with 1-Rs distinct and nearly half length of 1-M (short or absent in Pamphiliinae and Cephalciinae) (see Remarks to the subfamily Juralydinae above).

*Dolicholyda* gen. nov. differs from other Juralydinae in having pterostigma sclerotized only along sides (other genera have pterostigma fully sclerotized except *Juralyda*), cell 1r nearly twice as long as wide (other genera have the ratio less than 1.5 times except twice in *Tapholyda*) and cell 1mcu longer than pterostigma (other genera have 1mcu shorter, except nearly isometric in *Scabolyda latusa* Zhuang, Shih, Wang and Ren, 2022 and *S. tenuis* Zhuang, Shih, Wang and Ren, 2022, ([30], Figures 2F and 4C)). It differs from all genera but *Atocus* by 1-M vertical to 1-Cu (in *Tapholyda* these form an acute angle, in *Juralyda* and *Scabolyda*, an obtuse angle). It differs from *Juralyda* in 3r-m strongly oblique (almost vertical in *Juralyda*). In order to better distinguish the genera of Juralydinae, we establish the relevant key to the identification of different genera in Juralydinae.


**Key to the genera of Juralydinae**
1. Forewing with Sc not forked…………………………………………………………………………………2Forewing with Sc developed and forked……………………………………………………………………‥.32. Forewing with pterostigma only sclerotized along margins, 3r-m almost vertical……………………‥………………………………………………………………………………………………………………*Juralyda*Forewing with pterostigma completely sclerotized, 3r-m clearly oblique…………………………..*Atocus*3. Forewing with M+Cu smoothly bent medially…………………………………………………...*Tapholyda*Forewing with M+Cu angularly bent………………………………………………………………………….44. Forewing with pterostigma only sclerotized along margins, cell 1mcu obviously longer thanlength of pterostigma……………………………………………………………………*Dolicholyda* gen. nov.Forewing with pterostigma completely sclerotized, cell 1mcu subequal or shorter than length ofpterostigma…………………………………………….……………………………………………….*Scabolyda*

*Dolicholyda obtusata* Zhuang, Rasnitsyn, Shih and Wang, sp. nov.

Material. Holotype, CNU-HYM-ND2022105p/c (part and counterpart; Figure 1 and Figure 2) and paratype, CNU-HYM-ND2022106p/c (part and counterpart; Figure 3).

Etymology. The species name is a Latin word and means obtuse, referring to the obtuse apical part of cell a in hind wing.

Locality. Nanyingpan Village, Sandaogou Township, Duolun County, Inner Mongolia, China. Yixian Formation, Lower Cretaceous.

Diagnosis. In addition to the generic diagnosis, forewing with ap-Cu short; pterostigma slender and ca. 4.7 × as long as wide, its basal part blunt; Rs+M ca. 1.7–1.8 × as long as 2-M; 2m-cu angular medially. Hind wing with cell a wide and blunt apically; vein 2A sharply curved in the end.

Description. Holotype. Female. Head, thorax, and abdomen dark brown. Antenna with scape dark brown except the apical part drab; pedicel and possibly first flagellomere drab, other flagellomeres dark brown (Figure 1E).

Head flat and square, width slightly longer than length; clypeal margin slightly wavy; mandibles large and sharp; eyes medium-length. Antenna with 15~16 segments and approximately twice as long as width of head; scape at least twice as long as pedicel, and 1.5 × as wide as pedicel; pedicel 0.3 × as long, and 1.2 × as wide as first flagellomere; first flagellomere not subsegmented and nearly as long as length of next three flagellomeres combined, second flagellomere 2.7 times as long as wide and slightly shorter than third. Remaining flagellomeres gradually shortening.

Thorax without punctations. Pronotum slightly wider than the head (the other structures of thorax indistinct). Legs partly preserved and coxae inverted trapeziform. Abdomen with eight segments preserved, as wide as the thorax. Ovipositor with valvula 1 and valvula 2 partly visible (Figure 2A).

Forewing with pterostigma long and slender, sclerotized along margins, its basal part blunt, nearly 4.7 × as long as wide (Figure 1G,H). Sc bifid, Sc2 entering R before 1-Rs; R bent near the base of 1-Rs. 1-Rs inclined toward wing apex and nearly half the length of 1-M; 2-Rs ca. 0.6 × as long as Rs+M; 3-Rs strongly curved; 4-Rs almost half the length of 1-M. 2r-m located distad to 2r-rs and 1.3 × as long as 2-Rs. 3r-m inclined toward wing apex and meeting 5-Rs nearly at 116°. M+Cu bent sharply; ap-Cu very short and nearly as wide as the width of vein M+Cu (Figure 1C). Rs+M 1.7 × as long as 1-M; angle between 1-M and 1-Cu of 85°. Cell 1mcu nearly twice as long as wide, with cu-a located distad to the middle of the cell. 2-M 0.6 times as long as Rs+M; 3-Cu nearly 2.3 times as long as 1m-cu; 2m-cu curved near the middle. The base of vein 2A+3A having a protrusion at the bend close to vein 1A (Figure 1C). Cell 2a almost 3.5 times as long as wide.

Hind wing with Sc present and partly preserved (Figure 1F). 1-Rs nearly as long as 1-M; 1r-m located distad to the base of 1-Rs, and nearly aligned with 1-M; 3r-m inclined toward the wing apex, slightly sigmoidal; m-cu bent, located distad to the middle of cell rm, and separated from 3r-m by a distance of almost 1.2 × its length; cu-a proximad to the middle of cell mcu and meeting 1-Cu at the angle of 117°. 1A nearly straight; 2A sharply bent apically (Figure 1D). Cell a blunt apically with apex width nearly equal to length of 1-M.

Paratype (Figure 3). Sex unknown. Body surface without punctations. Head subcircular. Thorax slightly wider than the width of head. Legs preserved with basal parts. Abdomen with eight segments visible. The structure of genitalia not preserved.

Forewing with pterostigma slender and sclerotized along sides, nearly 4.7 × as long as wide (Figure 3G,H); Sc developed and forked, Sc1 entering C slightly distal to the base of 1-Rs. 1-Rs 0.4 times as long as 1-M and inclined toward wing apex. 2-Rs ca. 0.3 × as long as Rs+M. 4-Rs shorter than half of 1-M. 2r-m ca. 1.9 × as long as 2-Rs. Rs+M nearly 1.8 × as long as 2-M. Cell 1mcu longer than pterostigma and nearly twice as long as wide. Vein M+Cu bent, the extra stub ap-Cu extremely short (Figure 3F). 1-M meeting 1-Cu nearly at the angle of 94°. 3-Cu ca. 2.6 × as long as 1m-cu. Cell 2a nearly 3.3 × as long as wide.

In hind wing, cell c narrow in the basal part, slender fusiform. Sc developed and not forked, originating from the middle of vein R, and almost entering apex of vein C; its length occupying half of cell c (Figure 3C–E).

Dimensions (in mm). CNU-HYM-ND2022105p/c: body length (excluding antenna) 14.12; head width 3.23, length 2.95; the first flagellomere as preserved 1.16 in length; forewing length 11.81, width 4.86. CNU-HYM-ND2022106p/c: body length 12.73; head width 2.68, length 2.12; forewing length at least 8.11.

*Dolicholyda confluens* Zhuang, Rasnitsyn, Shih and Wang, sp. nov.

Material. Holotype, CNU-HYM-ND2022107p/c (part and counterpart; Figure 4).

Etymology. The species epithet is a Latin word meaning confluence. It refers to the extremely long ap-Cu joining Cu and 1A in the forewing.

Locality. Nanyingpan Village, Sandaogou Township, Duolun County, Inner Mongolia, China. Yixian Formation, Lower Cretaceous.

Diagnosis. *Dolicholyda confluens* sp. nov. can be easily separated from all other pamphiliids by the extremely long ap-Cu of the forewing, which joins vein 1A in the end. Additionally, the pterostigma is slender and ca. 4.8 × as long as wide; Rs+M ca. 1.5 × as long as 2-M; 2m-cu angular medially.

Description. Holotype (Figure 4). Sex unknown. The whole body dark brown except head light brown, and its surface lacking punctations (head and thorax damaged with distortion). Mesopseudosternum nearly triangular and large. Abdomen with eight segments, each segment short and wide. The widest segment 1.2 times as wide as thorax. Genitalia not preserved.

Forewing with pterostigma long and narrow, sclerotized around margins, nearly 4.8 × as long as wide (Figure 4E). Sc bifid, Sc2 joining R before the base of 1-Rs and obviously shorter than Sc1. R strongly bent at the confluence with Sc2. 1-Rs inclined toward wing apex, and ca. 0.6 × as long as 1-M; 3-Rs strongly arched; 4-Rs nearly half the length of 1-M. 2r-m parallel to 2r-rs and nearly twice as long as 2-Rs. 3r-m inclined toward wing apex. Cell 3rm as wide as cell 3r in its basal part. Rs+M 1.6 × as long as 1-M. Angle between 1-M and 1-Cu nearly of ca. 95°. M+Cu bending sharply distad to the middle part, ap-Cu stub extremely long and entering vein A anteriad crossvein a (Figure 4D). Rs+M ca. 1.5 × as long as 2-M. Cell 1mcu slightly longer than pterostigma and 1.7 × as long as wide. 3-Cu 2.7 × as long as 1m-cu; 2m-cu curved near its middle. Cell 2a almost 3.1 × as long as wide. Hind wing not preserved.

Dimensions (in mm). Holotype, CNU-HYM-ND2022107p/c: body length at least 11.87; head width 3.36; forewing length 13.00, width 5.60.

*Dolicholyda angusta* Zhuang, Rasnitsyn, Shih and Wang, sp. nov.

Material. Holotype, CNU-HYM-ND2022108 (Figure 5).

Etymology. The species name is a Latin word meaning narrow, referring to narrow cell a in hind wing.

Locality. Nanyingpan Village, Sandaogou Township, Duolun County, Inner Mongolia, China. Yixian Formation, Lower Cretaceous.

Diagnosis. In addition to the generic diagnosis, forewing with ap-Cu absent; pterostigma wide and ca. 3.5 × as long as wide, its basal part acute; Rs+M ca. 1.9 × as long as 2-M 2m-cu almost straight medially. Hind wing with cell a relatively narrow apically; vein 2A smoothly curved in the end.

Description. Holotype (Figure 5). Female. Body surface sepia and lacking punctations. Antennae partly preserved and dark brown.

Head rounded rectangular, transverse, with occipital carina apparently complete, with submedial branches toward occipital foramen, with structures delimited below occipital foramen possibly representing postgenae (lower part of postocciput) and lower tentorial bridge comparable to those in living Pamphiliidae ([31], Figure 136) except more narrow lower tentorial bridge (Figure 5E). Antennae obviously longer than the width of head, with at least sixteen segments; dark brown. Several terminal flagellomeres equal in lengths and widths, except apical flagellomere nearly twice as long as wide. Thorax poorly preserved. Legs short, poorly preserved. Abdomen with nine segments and the middle part inflated. Ovipositor very short, valvula 1 and valvula 2 widely separated basally and meeting only at the apex, valvula 3 located on both sides of valvula 2.

Forewing with pterostigma relatively wide, sclerotized at sides and nearly 3.5 × as long as wide, its basal part slightly acute. Sc bifid, Sc2 entering R anteriad base of 1-Rs. 1-Rs ca. 0.4 × as long as 1-M. 2-Rs nearly half length of Rs+M. 3-Rs curved, 4-Rs short and ca. 0.3 × as long as 1-M. 2r-m ca. 1.7 × as long as 2-Rs. 3r-m inclined toward apical part of wing. ap-Cu absent (Figure 5F). Angle between 1-M and 1-Cu nearly of 80°. Length proportions of vein 1-M: Rs+M: 1-Cu: 2-M = 1: 1.9: 1.7: 1; cu-a located slightly distad to the middle of cell 1mcu, 3-Cu ca. 2.5 × as long as 1m-cu. Cell 1mcu 2.2 × as long as wide and obviously longer than pterostigma. 2m-cu slightly curved. Cell 2a nearly 3.5 × as long as wide.

Hind wing with Sc developed, Sc extending out from the middle part of R and its length twice as long as 1-Rs (Figure 5G). 1-Rs ca. 2.5 × as long as 1r-m. 1-M nearly 1.9 × as long as 1r-m and angle between them of 144°; cu-a located at middle of cell mcu and angle between 1-Cu and cu-a nearly 100°. 1A gently curved throughout its length, 2A bent apically. Cell a narrow and its apical part acute.

Dimensions (in mm). Holotype: body length (excluding antenna) 9.95; head width 2.38, length 1.87; antenna at least 2.81 in length; forewing length 8.22, width 3.46.

Remarks. A small round structure clearly visible within the occipital foramen is difficult to interpret: it could be an antennal attachment orifice or, rather, simply a damage in the head cuticle.

In *Dolicholyda* gen. nov., we can easily distinguish species by the length of ap-Cu, which is absent in *D. angusta* sp. nov., short in *D. obtusata* sp. nov. and extremely long (even joins 1A) in *D. confluens* sp. nov. Leaving this feature aside, *D. obtusata* sp. nov. show much similarity with *D. confluens* sp. nov., especially on the characters of forewing with pterostigma slender and ca. 4.7 × as long as wide, its basal part blunt; 2m-cu angular medially. In contrast, in *D. angusta* sp. nov., pterostigma wide and ca. 3.5 × as long as wide, its basal part acute; 2m-cu almost straight medially. Additionally, *D. obtusata* sp. nov. also differs from *D. angusta* sp. nov. in hind wing with cell a blunt apically vs. acute in *D. angusta* sp. nov.

## 4. Discussion

The discovery of *Dolicholyda* gen. nov. in the Early Cretaceous Yixian Formation, NE China, sheds new light on the diversity and the characteristics of the subfamily Juralydinae and the family Pamphiliidae in general. The present study indicates wide morphological diversity of the subfamily, including the characters that were earlier treated as diagnostic features of the subfamily, such as length of cell 1mcu, presence or absence of angular bend of M+Cu, and the presence and length of ap-Cu at that bend. As a result, a long 1-Rs remains the only uniquely diagnostic character of Juralydinae.

The vein ap-Cu is present in most genera of extant Pamphiliidae while showing a wide diversity of variations in its length, except in *Acantholyda* with ap-Cu usually absent or extremely short [32]. However, among fossil pamphiliid taxa, most species lack ap-Cu or only possess an extremely short ap-Cu. Up to now, only *Atocus*, *Ulteramus* and our new genus *Dolicholyda* gen. nov. have ap-Cu (extremely short in *Atocus*). According to previous studies, the length change of ap-Cu is subject to intraspecific variation [18]. Shinohara [33] found that the length of ap-Cu was unstable in the same species of *Acantholyda*, sometimes absent or very short, and rarely long. But in general, variation is feeble. Our new genus also displayed varying lengths of ap-Cu in the Lower Cretaceous Pamphiliidae. Furthermore, *Dolicholyda confluens* has extremely long ap-Cu (Figure 4D,F), which is rare in fossils and there were no cases of ap-Cu joining 1A documented before. This unique case provides a new extreme value for the length variation of ap-Cu and indicates that ap-Cu absent or extremely short is not the stable character for Juralydinae.

Besides the stub of ap-Cu, there are also some bud-like protrusions at the bend of 2A+3A in the forewings of *D. obtusata* and *D. confluens* (Figure 1C and Figure 4F). Similarly, some other records about wing variations in fossil Symphyta have been also reported in the Mesozoic sawflies (mainly referring to supernumerary veins there), such as *Chionoxyela nivea* Rasnitsyn, 1993; *Shartexyela mongolica* Rasnitsyn, 2008; *Xyelocerus diaphanous* Gao, Ren and Shih, 2009; *Hoplitolyda duolunica* Gao, Shih, Rasnitsyn and Ren, 2013; *Potrerilloxyela menendez* Lara, Rasnitsyn and Zavattieri, 2014; *Aethotoma aninomorpha* Gao, Shih, Engel and Ren, 2016; *Pamparaphilius khasurtensis* Kopylov and Rasnitsyn, 2017; *Trematothorax extravenosus* Kopylov and Rasnitsyn, 2017; and *Brachyoxyela conjunctiva* Dai, Shih, and Rasnitsyn, 2022 [34,35,36,37,38,39,40,41,42]. The variation of their wing veins looks irregular. Extra veins appear transversal, longitudinal, or annular. They may appear anywhere on the forewing or hind wing, located close to the wing base, middle or apex (Figure 6). Judging from the species referred to above, these abnormalities are more likely to occur in the wing apex (six out of eight of them in the apical part). To a certain extent, it reflects the instability of wing venations in Symphyta. Interestingly, the variations in *Chionoxyela nivea* and *Hoplitolyda duolunica* are located at the bend of M+Cu, which is similar to that of Pamphiliidae, except that it is inverted in direction (i.e., directed toward the anterior margin of the wing; Figure 6C,D).

The presence of vein Sc in the hind wing is a feature inherited from a non-hymenopteran ancestor. According to the statistics of Vilhelmsen [1], only some of the primitive taxa currently have this feature, such as the Xyelidae and the Pamphiliidae. In more derived taxa, there are more records of Sc in the extinct fossil groups, such as Xyelydidae Rasnitsyn, 1968; Xyelotomidae Rasnitsyn, 1968; Siricidae Billberg, 1820 and Mirolydidae Wang, Rasnitsyn and Ren, 2017 [4,36,43]. Throughout the extant groups of Pamphiliidae, vein Sc is long and parallel to R (Figure 7A–D). Its anterior branch Sc1 joins C near the apex of cell c and the posterior branch Sc2 joins R at two-thirds of the length of vein R [32,44]. By contrast, in *Dolicholyda*, Sc is very short, not forked and slightly shorter than half length of cell c. It extends out from the middle part of R and enters C in its apical part (Figure 3C–E, Figure 5G and Figure 7F). In *Scabolyda*, Sc is appressed to R except distally (a double vein is maybe present, Figure 7E). In general, there are some differences between the Sc of extant Pamphiliidae and fossil genera (*Dolicholyda* and *Scabolyda*). However, its diagnostic value is obscure, and we need more specimens to define its value as a diagnostic feature.

## 5. Conclusions

*Dolicholyda obtusata* gen. et sp. nov., *D. confluens* sp. nov., and *D. angusta* sp. nov. are described based on four fossil specimens from the Lower Cretaceous Yixian Formation in Duolun County, Inner Mongolia, China. The new species display varying lengths of ap-Cu in their forewings and an unforked Sc in the hind wings. We treat the long 1-RS as the only valid diagnostic character of Juralydinae based on the new material described herein. These new findings enhance our understanding of the wing venation characteristics of fossil pamphiliids and expand the diversity of Pamphiliidae in the Mesozoic.

## Figures and Tables

**Figure 1 insects-13-00947-f001:**
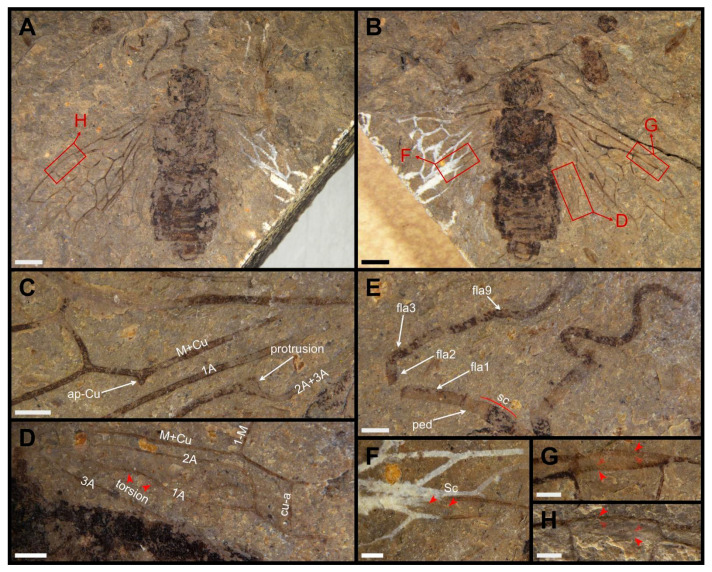
*Dolicholyda obtusata* gen. et sp. nov., Holotype (CNU-HYM-ND2022105p/c), female. (**A**), ventral view. (**B**), dorsal view. (**C**), ap-Cu and protrusion in forewing. (**D**), break and torsion in hind wing. (**E**), antennae. (**F**), Sc in hind wing. (**G**,**H**), pterostigma. Symbols: sc = scape; ped = pedicel; fla1, fla2, fla3, fla9 = 1st, 2nd, 3rd, and 9th flagellomere. Arrows in red mean the venations here are highlighted. Scale bars: 2 mm in (**A**,**B**); 0.5 mm in (**C**–**H**).

**Figure 2 insects-13-00947-f002:**
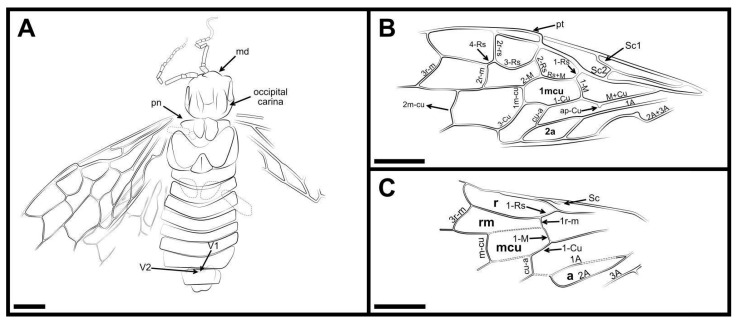
*Dolicholyda obtusata* gen. et sp. nov., (**A**), line drawing of ventral view. (**B**), line drawing of forewing. (**C**), Reconstruction of the hind wing venation (mirror flip for the anal field). Symbols: md = mandible; pn = pronotum; v1, v2 = 1st and 2nd valvifer; pt = pterostigma. Cell names are in bold. Scale bars: 2 mm in (**A**–**C**).

**Figure 3 insects-13-00947-f003:**
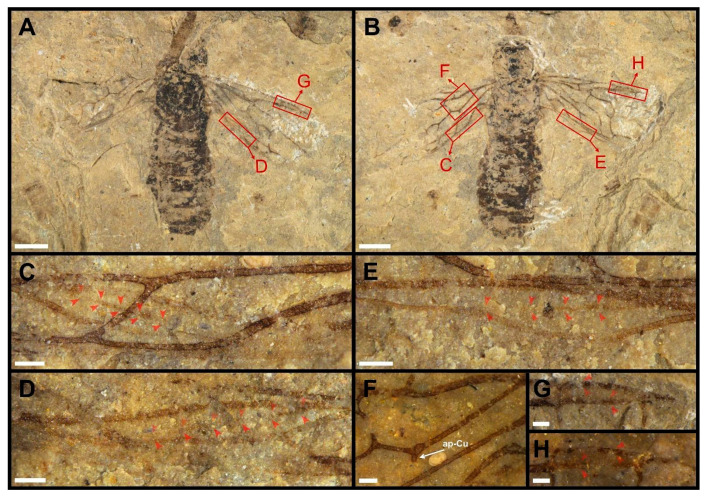
*Dolicholyda obtusata* gen. et sp. nov., Paratype (CNU-HYM-ND2022106p/c), sex unknown. (**A**), dorsal view. (**B**), ventral view. (**C**–**E**), Sc in hind wing. (**F**), ap-Cu in forewing under alcohol. (**G**), pterostigma. (**H**), pterostigma under alcohol. Arrows in red mean the venations here are highlighted. Scale bars: 2 mm in (**A**,**B**); 0.25 mm in (**C**–**H**).

**Figure 4 insects-13-00947-f004:**
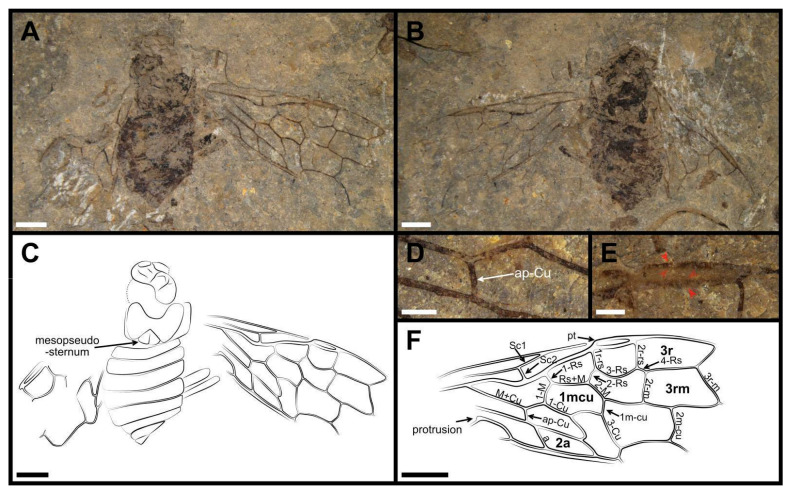
*Dolicholyda confluens* sp. nov., Holotype, (CNU-HYM-ND2022107p/c), sex unknown. (**A**), ventral view. (**B**), dorsal view. (**C**), line drawing of ventral view. (**D**), ap-Cu in forewing. (**E**), pterostigma. (**F**), line drawing of forewing. Arrows in red mean the venations here are highlighted. Cell names are in bold. Scale bars: 2 mm in (**A**–**C**,**F**); 0.5 mm in (**D**,**E**).

**Figure 5 insects-13-00947-f005:**
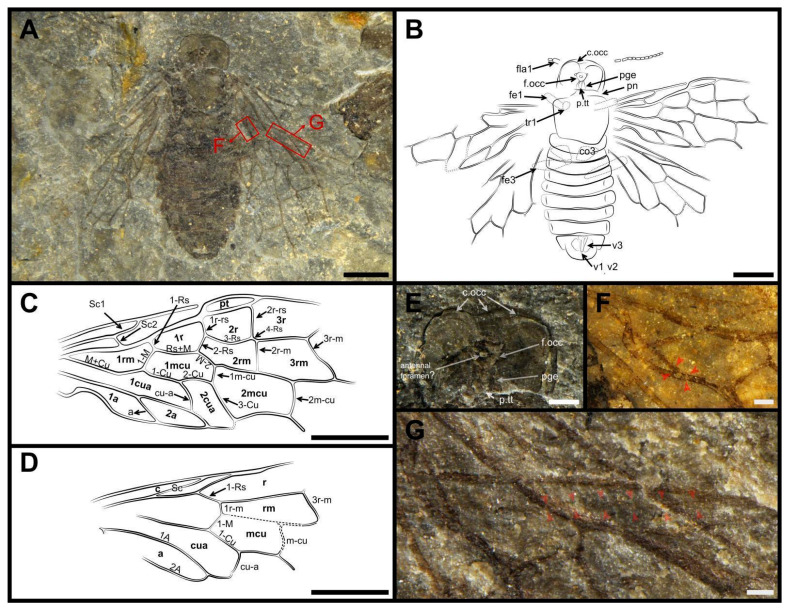
*Dolicholyda angusta* sp. nov., Holotype, (CNU-HYM-ND2022108), female. (**A**), dorsal view. (**B**), line drawing of dorsal view with forewing and hind wing artificially rotated from body and each other for better viewing. (**C**), line drawing of forewing after reconstruction (mirror flip for the anal field). (**D**), line drawing of hind wing. (**E**), head under alcohol. (**F**), vein M+Cu in forewing under alcohol. (**G**), Sc in hind wing. Symbols: fe1, fe3 = fore and hind femur; tr1 = fore trochanter; v3 = 3rd valvifer; co3 = hind coxa; c.occ = occipital carina; f.occ = occipital foramen; pge = postgena; p.tt = tentorial bridge. Cell names are in bold. Arrows in red mean the venations here are highlighted. Scale bars: 2 mm in (**A**–**D**); 0.5mm in (**E**); 0.25 mm in (**F**,**G**).

**Figure 6 insects-13-00947-f006:**
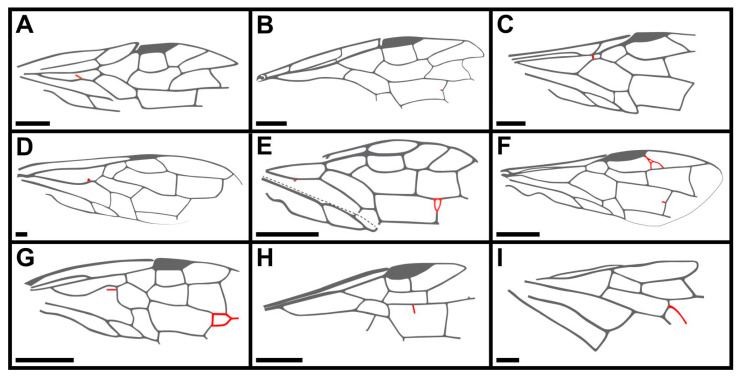
Several examples of wing venation variation in fossil Symphyta. (**A**), *Chionoxyela nivea* Rasnitsyn, 1993 (Xyelidae) ([34], Figure 12). (**B**), *Shartexyela mongolica* Rasnitsyn, 2008 (Xyelidae) ([35], Figure 1B). (**C**), *Xyelocerus diaphanous* Gao, Ren and Shih, 2009 (Xyelotomidae) ([36], Figure 2B). (**D**), *Hoplitolyda duolunica* Gao, Shih, Rasnitsyn and Ren, 2013 (Praesiricidae) ([37], Figure 3B). (**E**), *Potrerilloxyela menendez* Lara, Rasnitsyn and Zavattieri, 2014 (Xyelidae) ([38], Figure 4C). (**F**), *Aethotoma aninomorpha* Gao, Shih, Engel and Ren, 2016 (Xyelotomidae) ([39], Figure 2D). (**G**), *Pamparaphilius khasurtensis* Kopylov and Rasnitsyn, 2017 (Sepulcidae) ([40], Figure 1). (**H**), *Trematothorax extravenosus* Kopylov and Rasnitsyn, 2017 (Sepulcidae) ([41], Figure 5). (**I**), *Brachyoxyela conjunctiva* Dai, Shih and Rasnitsyn, 2022 (Xyelidae) ([42], Figure 3C). Scale bars: 2 mm in (**A**–**I**). Notes: specimens of *Hoplitolyda duolunica*, *Xyelocerus diaphanous* and *Aethotoma aninomorpha* have both wings preserved and only *Hoplitolyda duolunica* has symmetric venation variation. Other specimens have only a single wing preserved.

**Figure 7 insects-13-00947-f007:**
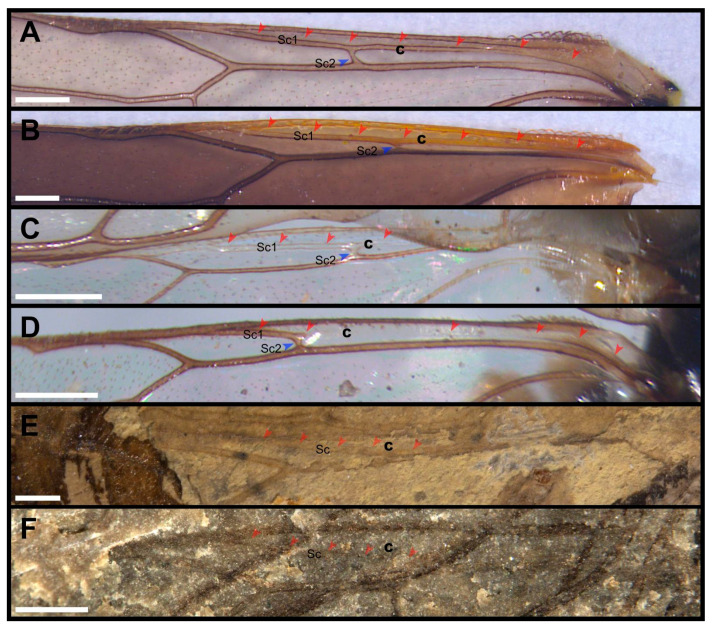
Sc in hind wings of Pamphiliidae. (**A**), *Acantholyda posticalis* Matsumura, 1912, ♀ (Cephalciinae). (**B**), *Cephalcia yanqingensis* Xiao, 1987, ♀ (Cephalciinae). (**C**), *Pamphilius ignymonriensis* Lacourt, 1973, ♂ (Pamphiliinae). (**D**), *Neurotoma sibirica* Gussakovskij, 1935, ♀ (Pamphiliinae). (**E**), *Scabolyda latusa* Zhuang, Shih, Wang and Ren, 2022, ♀ (Juralydinae). (**F**), *Dolicholyda angusta* sp. nov., ♀ (Juralydinae). Cell c are in bold. Arrows here mean the venations are highlighted. Scale bars: 0.5 mm in (**A**–**F**).

## Data Availability

All data from this study are available in this paper and the associated papers.

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
