# Peer review of "New Pamphiliids with Varying Venations from Lower Cretaceous Yixian Formation of Northeast China (Hymenoptera, Pamphiliidae)†"

_insects, 2022, doi:10.3390/insects13100947_

Round 1

Reviewer 1 Report

Zhuang et al describe a new genus and three new species of pamphiliid wasps from the Cretaceous of China. These additions are extremely valuable given the scarcity of the pamphiliid fossil record. In addition, they discuss the wing venation variability found in fossil and extant species and its potential diagnostic value. The illustrations are qualitative and the descriptions robust. I only have minor comments and corrections on this manuscript (indicated in the attached pdf), all will be easily implemented by the authors.

Author Response

Thank you for your kind letter and valuable comments. We have accepted and adopted nearly all comments and corrections, and revised our manuscript accordingly.

Key revisions and our replies to comments are as follows:

-Line 47 in the revised manuscript: Daohugou is late Middle Jurassic to early Late Jurassic age (e.g., Huang et al., 2018; Huang, 2019; Lian et al., 2021).

Reply: Thanks. Have corrected!

-For this section be sure that the subscript of the font (Sc1 and Sc2) match the nomenclature used on your drawings

the 1 and 2 are not treated in the same way as in your drawings

Reply: Thanks. Have corrected!

-For all your figure captions please indicate that the cell names are in bold

Reply: Thanks. Have corrected!

-Line 110: I suggest keeping a certain homogeneity in the diagnosis, with first head characters, then forewing. Here it's first wing, then the head, and the wing

Reply: Have corrected!

-Line 143: I don't think that this is a lack of color but I think this is rather a lack of sclerotization. The same character is found in Xyelidae.

Reply: Have corrected!

-Line 145: from what I can measure on your drawings Rs+M/2-M ratio ranges between 1.5 to ca. 1.9 so not exactly what you state in the diagnosis.

Please revised this character

See your fig 5 the 2-M is really short compared to that of fig 4

Reply: Have corrected!

-Line 176: I suggest adding "subequal" to match your comments in the remark section of the genus

Reply: Have corrected!

-Line 202: In all the sepecies diagnosis, I think you can use the Rs+M/2-M ratio of each species as a distinctive character because you have something like: 1.68, 1.5, 1.9. ;-)

Maybe the shape of the Sc1 is also something to add in each diagnosis, sometimes linear sometimes bent

Reply: Have corrected! We add the ratio of Rs+M/2-M.

-Line 215: I think this is rather a comment so I suggest placing this part ( The other structures of thorax indistinct.) in parentheses.

Reply: Have corrected!

-you have inverted 3-Cu and 1m-cu on your drawing. please correct

Reply: Thanks. Have corrected!

-not sure that this is needed, it is not a descriptive but rather interpretative

Reply: Have deleted!

-Line 244: for me the length of the vein is similar to the width of M+Cu (same condition as in the holotype

Reply: Have corrected!

-Line 268: At least add the state of the characters used for the diagnosis of the type species Dolicholyda confluens

Reply: Have corrected!

-Line 280: rather 2-M ? or another vein ?

Reply: refer to the angle between Rs+M and 1-M. Have deleted this sentence!

-Line 283: If 0.7 it means that Rs+M/2-M ratio is equal ca. 1.4 so slightly different from what I measure (ca. 1.5) please check again

Reply: Have corrected!

-Line 328: I think "curved" is more appropriate as there is no clear angle (compare to that of M+Cu for example)

Reply: Have corrected!

-add the label of this vein on the figure and be sure that all the veins mentioned in the three descriptions are labeled on your drawings please

Reply: Have added!

-Line 358: and Tapholyda caplani ?

Reply: We refer to the extant species here, so we do not add Tapholyda caplani. 

-Line 372: what do you mean ? It is not clear to me

Reply: Have corrected! We made some adjustments in this part.

Reviewer 2 Report

Dear Dr. Wang,

I have carefully read your manuscript entitled "New pamphiliids with varying venations from Lower Cretaceous Yixian Formation of Northeast China (Hymenoptera, Pamphiliidae)†". I believe this paper contains new important information on a new extinct genus and three new extinct species of the family Pamphiliidae from China, and therefore it could be published in Insects. A few corrections to the text are suggested in the attached file.

Best regards,

Author Response

Thank you for your kind letter and valuable comments. We have accepted and adopted all comments and corrections, and revised our manuscript accordingly.

Key revisions and our replies to comments are as follows:

-Siricidae is an extant family. Do you mean its extinct members?

Reply: Yes. Some extinct species of Siricidae, such as Gigasirex Rasnitsyn, 1968 and Cretosirex Wang et al., 2018, also have Sc in hind wing.